

# Delineating wetland catchments and modeling hydrologic connectivity using LiDAR data and aerial imagery

Qiusheng Wu[1], Charles R. Lane[2]

[1]Department of Geography, Binghamton University, State University of New York, Binghamton, NY 13902, USA

[2]U.S. Environmental Protection Agency, Office of Research and Development, National Exposure Research

Laboratory, 26 W. Martin Luther King Dr., Cincinnati, OH 45268, USA

*Correspondence to*: Qiusheng Wu (wqs@binghamton.edu)

**Abstract:** In traditional watershed delineation and topographic modeling, surface depressions are generally treated as spurious features and simply removed from a digital elevation model (DEM) to enforce flow continuity of water across the topographic surface to the watershed outlets. In reality, however, many depressions in the DEM are actual wetland landscape features that are seldom fully filled with water. For instance, wetland depressions in the Prairie Pothole Region (PPR) are seasonally to permanently flooded wetlands characterized by nested hierarchical structures with dynamic filling-spilling-merging surface-water hydrological processes. The objectives of this study were to delineate hierarchical wetland catchments and model their hydrologic connectivity using high-resolution LiDAR data and aerial imagery. The graph theory-based contour tree method was used to delineate the hierarchical wetland catchments and characterize their geometric and topological properties. Potential hydrologic connectivity between wetlands and streams were simulated using the least-cost path algorithm. The resulting flow network delineated putative temporary or seasonal flow paths connecting wetland depressions to each other or to the river network at scales finer than available through the National Hydrography Dataset. The results demonstrated that our proposed framework is promising for improving overland flow simulation and hydrologic connectivity analysis.

**Keywords:** hydrologic connectivity, prairie pothole, wetland depressions, geographically isolated wetlands, flow path, LiDAR

## 1 Introduction

The Prairie Pothole Region (PPR) of North America extends from the north-central United States to south-central Canada, encompassing a vast area of approximately 720,000 km². The landscape of the PPR is dotted with millions of wetland depressions formed by the glacial retreat that happened during the Pleistocene Epoch (Winter, 1989). The PPR is considered as one of the largest and highly productive wetland areas in the world as it serves as a primary breeding habitat for North America's waterfowl population (Keddy, 2010; Steen et al., 2014; Rover and Mushet, 2015). The wetland depressions, commonly known as potholes, possess important hydrological and ecological functions, such as providing critical habitat for many migrating and breeding waterbirds (Minke, 2009), acting as nutrient sinks (Oslund et al., 2010), and storing surface water that can attenuate peak runoff during a flood event (Huang et al., 2011b). The pothole size ranges from a relatively small area of less than 100 m² to as large as 30,000





m², with an estimated median size of 1600 m² (Zhang et al., 2009; Huang et al., 2011a). Most potholes have a water
depth of less than 1 m with varying water permanency, ranging from ephemeral to permanent (Sloan, 1972). Due to
their small size and shallow depth, these wetlands are highly sensitive to climate variability and are vulnerable to
ecological, hydrological, and anthropogenic changes. Wetland depressions have been extensively drained and filled
due to agricultural expansion, which is considered as the greatest source of wetland loss in the PPR (Johnston,
2013). In a report to the United States (U.S.) Congress on the status of wetland resources, Dahl (1990) estimated that
the coterminous U.S. lost more than 50 percent of their original wetlands over a period of 200 years between the
1780s and the 1980s. More recently, Dahl (2014) reported that the total wetland area in the PPR declined by
approximately 300 km² between 1997 and 2009. This represents an average annual net loss of 25 km². Regarding
the number of depressions, it was estimated that the wetland depressions declined by over 107,000 or four percent
between 1997 and 2009 (Dahl, 2014).

The extensive wetland drainage and removal have increased precipitation runoff into regional river basins,

which is partially responsible for the increasing frequency and intensity of flooding events in the PPR (Miller and
Nudds, 1996; Bengtson and Padmanabhan, 1999; Todhunter and Rundquist, 2004). Concerns over flooding along
rivers in the PPR have stimulated the development of hydrologic models to simulate the effects of depression
storage on peak river flows (Hubbard and Linder, 1986; Gleason et al., 2007; Gleason et al., 2008; Huang et al.,
2011b). Since most of these prairie wetlands do not have surface outlets or well-defined surface water connections,
they are generally considered as geographically isolated wetlands (GIWs) (Tiner, 2003; Cohen et al., 2016; Lane and
D'Amico, 2016). Recently, the U.S. Environmental Protection Agency conducted a comprehensive review of over
1350 peer-reviewed papers with the aim to synthesize existing scientific understanding of how wetlands and streams
affect the physical, chemical, and biological integrity of downstream waters (U.S. EPA, 2015). The report concludes
that additional research focused on the frequency, magnitude, timing, duration, and rate of fluxes from GIWs to
downstream waters is needed to better identify wetlands with functions that significantly affect other waters and
maintain the long-term sustainability and resiliency of valued water resources (Rains et al., 2016).

In addition to the comprehensive review by the U.S. EPA (2015), a number of recent studies focusing on the

hydrologic connectivity of prairie wetlands have been reported in the literature. For example, Chu (2015) proposed a
puddle-to-puddle modeling framework to delineate prairie wetlands and characterize their dynamic hydro-
topographic properties in the Cottonwood Lake area (2.55 km²) using a 10-m resolution digital elevation model
(DEM). Vanderhoof et al. (2016) examined the effects of wetland expansion and contraction on surface water
connectivity in the PPR using time series Landsat imagery. Ameli and Creed (2016) developed a physically-based
hydrologic model to characterize surface and groundwater hydrologic connectivity of prairie wetlands. In a
comprehensive overview of wetland hydrology in the PPR, Hayashi et al. (2016) highlighted that prairie wetlands
and catchments should be considered as highly integrated hydrological units because the existence of prairie
wetlands depends on lateral inputs of runoff water from their catchments in addition to direct precipitation. To our
knowledge, however, few studies on the hydrology of prairie wetlands have treated wetlands and catchments as
integrated hydrological units. Furthermore, high-resolution light detection and ranging (LiDAR) data have rarely





been used in broad-scale (e.g., basin- or subbasin-scale) studies to delineate wetland catchments and model wetland
connectivity in the PPR.

In this paper, we present a semi-automated framework for delineating nested hierarchical wetland

depressions and their corresponding catchments as well as simulating wetland connectivity using high-resolution
LiDAR data. The hierarchical structure of wetland depressions and catchments was identified and quantified using
the localized contour tree method (Wu et al., 2015). The potential hydrologic connectivity between wetlands and
streams was characterized using the least-cost path algorithm. The resulting flow network delineated putative
temporary or seasonal flow paths connecting wetland depressions to each other or to the river network at scales finer
than available through the National Hydrography Dataset. The results demonstrated that our proposed framework is
promising for improving overland flow simulation and hydrologic connectivity analysis, which subsequently may
improve the understanding of wetland hydrological dynamics at watershed scales.
**2 Study area and datasets**
**2.1 Study area**
Our study area focused on the Pipestem River subbasin in the Prairie Pothole Region of North Dakota (Fig. 1). The
subbasin is an 8-digit Hydrologic Unit Code (#10160002) with a total area of approximately 2,770 km$^2$, covering
four counties in North Dakota (see Fig. 1). The climate of the subbasin is characterized by long, cold, dry winters
and short, mild, variably wet summers (Winter and Rosenberry, 1995). Average annual precipitation is
approximately 440 mm with substantial seasonal and annual variations (Huang et al., 2011a). The land cover of the
Pipestem subbasin is dominated by cultivated crops (44.1%), herbaceous vegetation (25.9%), and pay/pasture
(13.1%), with a substantial amount of open water (7.1%) and emergent herbaceous wetlands (5.6%) (Jin et al.,
2013). The Cottonwood Lake area (see the blue rectangle in Fig. 1), a long-term field research site established by the
U.S. Geological Survey (USGS) and the U.S. Fish and Wildlife Service (USFWS) in 1977 for wetland ecosystem
monitoring, has been a very active area of research for several decades (e.g., Sloan, 1972; Winter and Rosenberry,
1995; Huang et al., 2011a; Mushet and Euliss, 2012; Hayashi et al., 2016).
**2.2 LiDAR data**
The LiDAR elevation data for the Pipestem subbasin were collected in the late October of 2011 and distributed
through the North Dakota GIS Hub Data Portal (https://gis.nd.gov/, accessed December 30, 2016). The bare-earth
digital elevation models (DEMs) derived from LiDAR point clouds are freely available as 1-m resolution image tiles
(2 km × 2 km). The vertical accuracy of the LiDAR DEM is15.0 cm. In total, the Pipestem Subbasin consists of 786
DEM tiles with an aggregated file size of 22.66 GB. We created a seamless LiDAR DEM (see Fig. 1) for the
Pipestem subbasin by mosaicking 786 DEM tiles and used it for all subsequent data analyses. The elevation of the
subbasin ranges from 422 m to 666 m, with relatively high-elevation areas in the west and low-elevation areas in the
east.





The LiDAR intensity data for the Pipestem subbasin were also collected at 1-m resolution coincident with
the LiDAR elevation data collection. In general, the return signal intensities of water areas are relatively weak due to
water absorption of the near-infrared spectrum (Lang and McCarty, 2009; McCauley and Anteau, 2014). As a result,
waterbodies typically appear as dark features whereas non-water areas appear as relatively bright features in the
LiDAR intensity image. Thresholding techniques have been commonly used to distinguish water pixels from non-
water pixels (Huang et al., 2011b; Huang et al., 2014; Wu and Lane, 2016). In this study, the LiDAR intensity data
were primarily used to extract standing-water areas (i.e., inundation areas) while the LiDAR DEMs were used to
derive nested wetland depressions and their corresponding catchments above the standing-water surface.

### 2.3 Ancillary data

In addition to the LiDAR datasets, we used three ancillary datasets, including the 1-m resolution aerial imagery from
the National Agriculture Imagery Program (NAIP) of the U.S. Department of Agriculture (USDA), National
Wetlands Inventory (NWI) from the USFWS, and National Hydrography Dataset (NHD) from the USGS.
The NAIP imagery products were also acquired from the North Dakota GIS Hub Data Portal. The default
spectral resolution of the NAIP imagery in North Dakota is natural color (Red, Green, and Blue, or RGB).
Beginning in 2007, however, the state has been delivered with four bands of data: RGB and Near Infrared. We
downloaded and processed six years of NAIP imagery for the Pipestem subbasin, including 2003, 2004, 2006, 2009,
2012, and 2014. A small portion of the study area with the NAIP imagery is shown in Fig. 2. These time-series
NAIP imagery clearly demonstrate the dynamic nature of prairie pothole wetlands under various dry and wet
conditions. In particular, the extremely wet year of 2014 resulted in many individual wetlands to coalesce and form
larger wetland complexes (see the yellow arrows in Fig. 2). It should be noted that all the NAIP imagery were
collected during the summer growing season of agricultural crops. Since no coincident aerial photographs were
collected during the LiDAR data acquisition campaign in 2011, these NAIP imagery can serve as valuable data
sources for validating the LiDAR-derived wetlands catchments and hydrological pathways in this study.
The NWI data for our study area were downloaded from https://www.fws.gov/wetlands/ (accessed
December 30, 2016). These wetlands inventory data in this region were created by manually interpreting aerial
photographs acquired in the 1980s with additional support from soil surveys and field checking (Cowardin et al.,
1979; Huang et al., 2011b; Wu and Lane, 2016). Tiner (1997) reported that the target mapping unit, the size class of
the smallest group of NWI wetlands that can be consistently mapped, was between 1000 $m^2$ and 4000 $m^2$ in the
Prairie Pothole Region. It should be noted that the target mapping unit is not the minimum wetland size of the NWI.
In fact, there are a considerable amount of NWI wetland polygons smaller than the target mapping unit (1000 $m^2$). In
this study, we focused on the prairie wetlands that are greater than 500 $m^2$. Therefore, 5644 small NWI wetland
polygons (< 500 $m^2$) were eliminated from further analysis. In total, there were 32,016 NWI wetland polygons ($\geq$
500 $m^2$) across the Pipestem subbasin (Table 1). The total size of these NWI wetlands was approximately 279.5 $km^2$,
covering 10.1% of the Pipestem subbasin. The areal composition of NWI wetlands were freshwater emergent
wetlands (86.5%), lakes (7.5%), freshwater ponds (5.3%), freshwater forested/shrub wetland (0.4%), and riverine
systems (0.3%). The median size of wetlands ($\geq$ 500 $m^2$) in our study area was 1778 $m^2$. Although the NWI data is



the only spatially comprehensive wetland inventory for our study area, it is now considerably out-of-date, as it was
developed 30 years ago and it does not reflect the wetland temporal change (Johnston, 2013). The wetland extent
and type for many wetland patches have changed since its original delineation (e.g., Fig. 2). Nevertheless, NWI does
provide valuable information about wetland locations (Tiner, 1997; Huang et al., 2011b). In our study, the NWI
polygons were primarily used to compare with the wetland depressions delineated from the LiDAR DEM.
The NHD data were downloaded from http://nhd.usgs.gov (accessed December 30, 2016). There were 1840
polyline features in the NHD flowline layer for the Pipestem subbasin, with a total length of 1402.2 km and an
average length of 762.1 m. The NHD flowlines overlaid on top of the LiDAR DEM with is shown in Fig. 1. It is
worth noting that the majority of the NHD flowline features were found in the low-elevation areas in the east. The
high-elevation areas in the west where most NWI wetland polygons are located have very few NHD flowlines,
except for the Little Pipestem Creek. This implies that a large number of temporary and seasonal flow paths were
not captured in the NHD dataset. It is also worth noting that the NHD does not try to systematically measure stream
lines <1.6 km (Stanislawski, 2009; Lane and D'Amico, 2016). In this study, the NHD flowlines were used to
compare the LiDAR-derived flow paths using our proposed methodology.
**3 Methodology**
**3.1 Outline**
Our methodology for delineating nested wetland catchments and flow paths is a semi-automated approach consisting
of several key steps: (a) extraction of hierarchical wetland depressions using the localized contour tree method (Wu
et al., 2015); (b) delineation of nested wetland catchments; (c) calculation of potential water storage; and (d)
derivation of flow paths using the least-cost path search algorithm. The LiDAR DEM is used to delineate
hierarchical wetland depressions and nested wetland catchments. The LiDAR intensity imagery is used to extract
wetland inundation areas. The potential water storage of each individual wetland depression is calculated as the
volume between the standing water surface and the maximum water boundary where water may overspill into
downstream wetlands or waters. The flow paths representing surface water connectivity can then be derived
according to the potential water storage and simulated rainfall intensity. The flowchart in Fig. 3 shows the detailed
procedures of the methodology for delineating wetland catchments and flow paths.
**3.2 Extraction of hierarchical wetland depressions**
The fill-and-spill hydrology of prairie wetland depressions have received considerable attention in recent years
(Shaw et al., 2012; Shaw et al., 2013; Golden et al., 2014; Chu, 2015; Hayashi et al., 2016; Wu and Lane, 2016). It
is generally acknowledged that the fill-and-spill mechanism of wetland depressions results in intermittent hydrologic
connectivity between wetlands in the Prairie Pothole Region of North America. In this study, wetland depressions
were categorized into two groups based on their hierarchical structure: simple depressions and composite
depressions. A simple depression is a depression that does not have any other depressions embedded in it, whereas a
composite depression is composed of two or more simple depressions (Wu and Lane, 2016). As shown in Fig. 4(a),





for example, depressions A, B, C, D and E are all simple depressions. As water level gradually increases in these
simple depressions, they will eventually begin to spill and merge to form composite depressions. For instance, the
two adjacent simple depressions A and B can form a composite depression F (see Fig. 4(b)). Continuously,
composite depression F and simple depression C can further coalesce to form an even larger composite depression
G. Similarly, the two adjacent simple depressions D and E can coalesce to form a composite depression H.
It is worth noting that the flow direction of surface waters resulting from the fill-and-spill mechanism
between adjacent wetland depressions can be bidirectional, depending on the antecedent water level and potential
water storage capability of the depressions. Most previous studies simply assumed that water always flows
unidirectionally from an upper waterbody to a lower one. This assumption, however, does not apply when two
adjacent depressions share the same spilling elevation or when there is a groundwater hydraulic head preventing the
flow from one to another. For example, in Fig. 4(a), the water flow direction resulting from fill-and-spill between
depressions A and B can be bidirectional. If depression B fills up more quickly than depression A, then water will
flow from depression B to depression A through the spilling point, and vice versa. Depression with a high elevation
of antecedent water level does not necessarily spill to an adjacent depression with a lower elevation of antecedent
water level. The key factors affecting the initialization of spilling process leading to flow direction are the
depression ponding time and catchment precipitation conditions. If the rain or runoff comes from the east and that is
where depression B is, then it might fill more quickly than if the runoff comes from the west where depression A is.
The wetland depression whichever takes less time to fill up will spill to the adjacent depression and eventually
coalesce to form a larger composite depression. If no adjacent depression with the same spilling elevation is
available, the upstream wetland depression will directly spill to downstream wetlands or river streams. For example,
the largest fully-filled composite depression G will spill to the simple depression D or the composite depression H, if
available.
To identify and delineate the nested hierarchical structure of potential wetland depressions, we utilized the
localized contour tree method proposed by Wu et al. (2015). The concept of contour tree was initially proposed to
extract key topographic features (e.g., peaks, pits, ravines, and ridges) from contour maps (Kweon and Kanade,
1994). The contour tree is a tree data structure that can represent the nesting of contour lines on a continuous
topographic surface. Wu et al. (2015) improved and implemented the contour tree algorithm, making it a locally
adaptive version. In other words, the localized contour tree algorithm builds a series of trees rather than a single
global contour tree for the entire area. Each localized contour tree represents one disjointed depression (simple or
composite), and the number of trees represents the total number of disjointed depressions for the entire area. When a
disjointed depression is fully flooded, the water in it will spill to the downstream wetlands or waters through
overland flow. For example, Fig. 4(c) and (d) show the corresponding contour tree graphs for the composite
depressions in Fig. 4(b). Once the composition G is fully filled, water will spill into simple depression D or
composite depression H.

**3.3 Delineation of nested wetland catchments**



After the identification and extraction of hierarchical wetland depressions from the contour maps, various
hydrologically relevant terrain attributes can be derived based on the DEM, including flow direction, flow
accumulation, catchment boundary, flow path, flow length, etc. The calculation of flow direction is essential in
hydrological analysis because it frequently serves as the first step to derive other hydrologically important terrain
attributes. On a topographic surface represented in a DEM, flow direction is the direction of flow from each grid cell
to its steepest downslope neighbor. One of the widely used flow direction algorithms is the eight-direction flow
model known as the D8 algorithm (O'Callaghan and Mark, 1984), which is available in most GIS software packages.
Flow accumulation is computed based on flow direction. Each cell value in the flow accumulation raster represents
the number upslope cells that flow into it. In general, cells with high flow accumulation values correspond to areas
of concentrated flow (e.g. stream channels), while cells with a flow accumulation value of zero correspond to the
pattern of ridges (Zhu, 2016). Therefore, flow accumulation provides a basis for identifying ridgelines and
delineating catchment boundaries.
A catchment is the upslope area that drains water to a common outlet. It is also known as the watershed,
drainage basin, or contributing area. Catchment boundaries can be delineated from a DEM by identifying ridgelines
between catchments based on a specific set of catchment outlets (i.e., spilling points). In traditional hydrological
modeling, topographic depressions are commonly treated as spurious depressions (or is it "features") and simply
removed to create a hydrologically correct DEM, which enforces water to flow continuously across the landscape to
the catchment outlets (e.g., stream gauges, dams). In the PPR, however, most topographic depressions in the DEM
are real features that represent wetland depressions, which are rarely under fully-filled condition (see Hayashi et al.,
2016; Lane and D'Amico, 2016; Vanderhoof et al., 2016). As illustrated above, we use the localized contour tree
algorithm to delineate the hierarchical wetland depressions, which can be used as the source locations for delineating
wetland catchments. Each wetland depression (simple or composite) has a corresponding wetland catchment. As
shown in Fig. 4(b), the corresponding wetland catchment of each wetland depression is bounded by the vertical lines
surrounding that depression. For example, the wetland catchment of simple depression A is $Catchment_{lm}$, and the
wetland catchment of simple depression B is $Catchment_{mn}$. Similarly, the wetland catchment of composite
depression F is $Catchment_{ln}$, which is an aggregated area of $Catchment_{lm}$ and $Catchment_{mn}$, resulting from the
coalesce from simple depressions A and B.
**3.4 Calculation of water storage and ponding time**
The potential water storage capacity ($V$ [m$^3$]) of each wetland depression can be computed through statistical
analysis of the grid cells that fall within the depression (Wu and Lane, 2016):
$$V = \sum_{i=1}^{n} (C - Z_i) \cdot R^2 \quad (1)$$





where $C$ is the spilling elevation (m), i.e., the elevation of the grid cell where water spills out of the depression; $Z_i$
is the elevation of the grid cell $i$ (m); $R$ is the spatial resolution (m); and $n$ is the total number of grid cells that fall
within the depression.

The ponding time of a depression can be calculated as follows:

$$T = V / (A_c \cdot I) \cdot 1000 \quad (2)$$
where $V$ is the potential water storage capacity of the depression (m³); $A_c$ is the catchment area of the
corresponding depression (m²); and $I$ is the rainfall intensity (mm/h). For the sake of simplicity, we assume that the
rainfall is temporally and spatially consistent and uniformly distributed throughout the landscape and all surfaces are
impervious.

The proportion of wetland depression area ( $A_w$ ) to catchment area ( $A_c$ ) is calculated by:

$$P_{wc} = A_w / A_c \quad (3)$$
The wetland depression area ( $A_w$ ) refers to the maximum ponding extent of the depression. The proportion ( $P_{wc}$ )
can serve as a good indicator for percent inundation of the study area under extremely wet conditions (e.g.,
Vanderhoof et al., 2016).
**3.5 Derivation of surface-water flow paths**
Based on the computed ponding time of each depression under a specific rainfall intensity, the most probable
sequence of the overland flow path can be constructed. The depression with the least ponding time will first fill and
start to overspill down-gradient. In hydrology, the path which water takes to travel from the spilling point to the
downstream surface outlet or channel is commonly known as flow path. The distance it takes for water to travel is
known as flow length. In this study, we adopted and adapted the least-cost path search algorithm (Wang and Liu,
2006; Metz et al., 2011; Stein et al., 2011) to derive the potential flow paths. The least cost path algorithm requires
two input datasets: the DEM and the depression polygons. Given the fact that topographic depressions in high-
resolution LiDAR DEM are frequently a combination of artifacts and actual landscape features (Lindsay and Creed,
2006), the user can set a minimum size threshold for depressions to be treated as actual landscape features. In other
words, depressions with a size smaller than the threshold will be treated as artifacts, and thus removed from the
DEM. This results in a partially-filled DEM in which depressions smaller than the chosen threshold are filled to
enforce hydrologic flow while larger depressions are kept for further analysis. Based on the partially-filled DEM,
flow direction for each grid cell can be calculated using the D8 flow direction algorithm (O'Callaghan and Mark,
1984). The least cost path minimizes the cumulative cost (i.e., elevation) along its length. Flow paths are computed
by tracing down gradient, from higher to lower cells, following assigned flow directions. With the simulated
overland flow path, flow length can be calculated, which is defined as the distance between the spilling point of an
upslope wetland and the inlet of a downslope wetland or stream. In our study, hydrologic connectivity refers to the
water movement between wetland-wetland and wetland-stream via hydrologic pathways of surface water.





**3.6 Wetland Hydrology Analyst**
To facilitate automated delineation of wetland catchments and flow paths, we have implemented the proposed
framework as an ArcGIS toolbox – Wetland Hydrology Analyst, which is freely available for download at
https://GISTools.github.io/ (accessed December 30, 2016). The core algorithms of the toolbox were implemented
using the Python programming language. The toolbox consists of three tools: Wetland Depression Tool, Wetland
Catchment Tool, and Flow Path Tool. The Wetland Depression Tool asks the user to select a DEM grid, and then
executes the localized contour tree algorithm with user-defined parameters (e.g., base contour elevation, contour
interval, min. depression size, min. ponding depth) automatically to delineate hierarchical wetland depressions. The
depressional wetland polygons can be stored as ESRI Shapefiles or a Feature Dataset in a Geodatabase. Various
morphometric properties (e.g., width, length, size, perimeter, max. depth, mean depth, volume, elongatedness,
compactness) are computed and included in the attribute table of the wetland polygon layers. The Wetland
Catchment Tool uses the DEM grid and the wetland polygon layers resulted from the Wetland Depression Tool as
input, and exports wetland catchment layers in both vector and raster format. The Flow Path Tool can be used to
derive overland flow path of surface water based on the DEM grid and the wetland polygon layers.
**4 Results**
**4.1 Inundation mapping**
The LiDAR intensity image was primarily used to map inundation areas. Before inundation mapping, we applied a
median filter to smooth the LiDAR intensity image. The median filter is considered as an edge-preserving filter that
can effectively remove data noise while preserving boundaries between image objects (Wu et al., 2014).
Subsequently, a simple thresholding method was used to separate inundated and non-inundated classes. Similar
thresholding techniques have been used in previous studies to extract water areas from LiDAR intensity imagery
(Lang and McCarty, 2009; Huang et al., 2011b). By examining typical inundation areas and the histogram of the
LiDAR intensity imagery used in our study, we chose an intensity threshold value of 20. Grid cells with an intensity
value between 0 and 20 were classified as an inundated class while grid cells with an intensity value greater than 20
as a non-inundated class, which resulted in a binary image. In the binary image, each region composed of inundated
pixels that were spatially connected (8-neighbor) was referred to as a potential inundation object. The "boundary
clean" and "region group" functions in ArcGIS Spatial Analyst were then used to clean ragged edges of the potential
inundation objects and assign a unique number to each object. It should be noted that water and live trees might both
appear as dark features in the LiDAR intensity imagery and have similar intensity values, although trees are not
particularly common in this region. As a result, some trees were misclassified as inundation objects. To correct the
misclassifications and obtain reliable inundation objects, we further refined the potential inundation objects using
additional criteria with the aid of the LiDAR DEM. First of all, we assumed that each inundation object must occur
within a topographic depression in order to retain water. In other words, all inundation objects must intersect with
depression objects derived using the "sink" function in ArcGIS Spatial Analyst. Secondly, given the relatively flat
and level surface of inundated regions, the standard deviation of pixel elevations within the same inundation object



should be very small. By examining the standard deviation of pixel elevations of some typical inundation objects
and tree objects, we chose a threshold of 0.25 m, which is slightly larger than the vertical accuracy of the LiDAR
data (0.15 m). This step can be achieved using the "zonal statistics as table" in ArcGIS Spatial Analyst. Thirdly, we
only focused on wetlands greater than 500 m$^2$. Therefore, inundation objects with areas smaller than 500 m$^2$ were
eliminated from further analysis.
Using the above procedures, we identified 15,784 inundation objects (i.e., depressions ≥ 500 m$^2$ with water
as determined through LiDAR-based analyses), which were then compared against the NWI wetland polygons in
our study area. We have made the inundation map publicly available at https://GISTools.github.io/ (accessed
December 30, 2016). The identified inundation objects encompassed an area of approximately 278.5 km$^2$,
accounting for 10.1 % of the Pipestem subbasin. Using the empirical area-to-volume equation developed for this
region of the PPR (see Gleason et al., 2007; Wu and Lane, 2016), we estimated that the 15,784 inundated
depressions stored approximately 448.5 million m$^3$ of water. The histogram of inundation polygons is shown in Fig.
5(a). The median size of the inundation polygons identified using the LiDAR intensity data was 1828 m$^2$, which was
slightly larger than the reported median size of NWI polygons (Table 2). Surprisingly, 18,957 out of 32,016 NWI
wetland polygons did not intersect with the inundation objects. In other words, 59.2% of the NWI wetland polygons
mapped in the 1980s were found to be partly or completely dried out or destroyed during the LiDAR collection
period. The total area of these dried NWI wetlands were 43.6 km$^2$, accounting for 15.6% of the original NWI
wetland areas (279.5 km$^2$). The histogram of the dried NWI wetlands is shown in Fig. 5(b). It is worth noting that
most dried NWI wetlands were relatively small with a median size of 1212 m$^2$ (Table 2). The LiDAR intensity data
were acquired in late October 2011, an extremely wet month according to the Palmer Hydrological Drought Index
(Fig. 6). During this wet season, most wetlands would be expected to have abundant standing water. If no standing
water could be detected in a wetland patch during this extremely wet period, we can safely conclude that the wetland
patch had probably dried out during the past decades, although we could not infer the exact time when it occurred.
The 'dried' NWI wetlands could also be attributed to the source of error in the original NWI data, which has a
minimum mapping unit (i.e., the minimum sized wetland that can be consistently mapped) of 0.1 ha for the PPR
(Tiner, 1997). Figure 5(b) shows that 37% of the 'dried' NWI polygons are smaller than the minimum mapping unit
(1000 m$^2$). This implies that these small 'dried' NWI polygons could be due to the NWI mapping error. Figure 7
illustrates the difference in shape and extent between the LiDAR-derived wetland inundation maps and the NWI
wetland polygons. The areas of disagreement (discrepancy) can be partly explained by the different image
acquisition dates. As mentioned earlier, the NWI maps for Pipestem subbasin of the PPR were created in the early
1980s while the LiDAR data were acquired in 2011. Clearly, most small NWI wetlands (see blue-outline polygons
in Fig. 7) appeared to not have visible standing water. Conversely, large NWI wetlands exhibited expansion and
coalesced to form even large wetland complexes (see yellow-outline polygons in Fig. 7).
**4.2 Nested wetland depressions and catchments**
We applied the localized contour method on the LiDAR-derived DEM and identified 33,241 wetland depressions. It
should be noted that the 'wetland depression' refers to the maximum potential ponding extent of the depression. The



inundated wetland depressions identified in the prior section can be seen as a subset of these depressions with water
in them. The total area of the identified wetland depressions was approximately 554.5 km$^2$ (Table 3), accounting for
20% of the entire study area. This histogram of the wetland depressions is shown in Fig. 8(a). The median size of
wetland depressions was 2592 m$^2$, which is larger than that of the NWI wetland polygons as well as the inundation
polygons (see Table 2). Using Eq. (1), we estimated that the potential water storage capacity of the Pipestem
subbasin resulting from these wetland depressions is 782.8 million m$^3$, which is 1.75 times as large as the estimated
existing water storage (448.5 million m$^3$) for the 15,784 inundated wetlands mentioned above. As noted by Hayashi
et al. (2016), wetlands and catchments are highly correlated and should be considered as integrated hydrological
units. The water input of each wetland largely depends on runoff from the upland areas within the catchment. Using
the method described in Section 3.3, we delineated the associated wetland catchments for each of the 33,241
wetland depressions. The histogram of the delineated wetland catchments is shown in Fig. 8(b). The median size of
wetland catchments was 25,780 m$^2$, which is approximately ten times larger than that of the wetland depressions
(Table 3).
Using Eq. (3), we calculated the proportion of depression area to catchment area ($A_w / A_c$) for each wetland
depression. It was found that the proportion ranged from 0.04% to 83.72%, with a median of 14.31% (Table 3). Our
findings are in general agreement with previous studies (Hayashi et al., 2016). For instance, Hayashi et al. (1998)
reported an average proportion ($A_w / A_c$) of 9% for 12 prairie wetlands in the Canadian portion of the PPR.
Similarly, Watmough and Schmoll (2007) analyzed 13 wetlands in the Cottonwood Lake Area during the high-stage
period and reported an average proportion ($A_w / A_c$) of 18%. It should be noted that the average proportion of
wetland area to catchment area ($A_w / A_c$) reported in the above studies were calculated on the basis of a limited
number of wetlands. On the contrary, our results were computed from more than 30,000 wetland depressions and
catchments, which provides a statistically reliable result due to a much larger sample size.
**4.3 Flow paths and connectivity lengths**
Based on the LiDAR DEM and wetland depression polygon layer, we derived the complete flow path network for
our study area using the least-cost path algorithm. We have made the interactive map of hydrologic connectivity in
the Pipestem subbasin publicly available at https://GISTools.github.io/ (accessed December 30, 2016). A number of
data layers derived from our study are available on the map, such as the inundation polygons, wetland depressions,
wetland catchments, and flow paths. NWI polygons, NHD flowlines, LiDAR intensity image, LiDAR shaded relief,
and time-series aerial photographs are also available for results comparison and visualization. A small proportion of
the map is shown in Fig. 9. Clearly, the derived flow paths not only captured the permanent surface water flow paths
(see the thick blue NHD flowline in Fig. 9), but also the intermittent and infrequent flow paths that have not been
mapped previously. By examining the potential flow paths overlaid on the color infrared aerial photograph (Fig.
9(b)), we can see that the majority of flow paths appeared to be collocated with vegetated areas. This indicates that
flow paths are likely located in high soil moisture areas that are directly or indirectly related to surface water or
groundwater connectivity.





In total, there are 1840 NHD flowlines in the Pipestem subbasin. The mean and median length of NHD
flowlines are 762 m and 316 m, respectively (Table 4). However, the flow lengths derived from our study, which
connected not only stream segments but also wetlands to wetlands, revealed much shorter flow paths than the NHD
flowlines. This finding is within our expectation. The histogram of the derived flow lengths is shown in Fig. 10. The
median flow length is 83 m, which is approximately 1/4 of the median NHD flowlines. The median elevation
difference between an upstream wetland and a downstream wetland connected through the flow path is 0.89 m.
**5 Discussion**
It should be noted that the LiDAR data we used in this study were collected in the late October of 2011, which was
an extremely wet period according to the Palmer Hydrological Drought Index (see Fig. 6). During such wet period,
most wetlands exhibited high water levels and large water extents, which can be evidenced from the LiDAR
intensity image in Fig. 7 and the aerial photograph in Fig. 9. It can be clearly seen that most wetlands, particularly
those larger ones, appeared to have larger water extents compared to the NWI polygons. A substantial number of
NWI wetlands were found inundated to coalesce with adjacent wetlands and form larger wetland complexes. LiDAR
data acquired during high water levels is desirable for studying maximum water extents of prairie wetlands.
However, the use of wet-period LiDAR data alone is not ideal for studying the fill-and-spill hydrology of prairie
wetlands. Since LiDAR sensors working in the near-infrared spectrum typically could not penetrate water, it is
impractical to derive bathymetric information of the depression. As a result, the delineation and characterization of
individual wetland depressions nested within larger inundated wetland complexes were not possible. Bathymetric
LiDAR systems with a green laser onboard offer a promising solution for acquiring wetland basin morphometry due
to the higher penetration capability of the green laser (Wang and Philpot, 2007). In addition, the derivation of
antecedent water depth and volume of wetland depressions is difficult, which can only be estimated using empirical
equations based on the statistical relationship between depression area and depression volume (Hayashi and Van der
Kamp, 2000; Gleason et al., 2007). As noted earlier, the volume of water in the 15,784 inundated wetlands was
estimated to be 448.5 million $m^3$. Ideally, using multiple LiDAR datasets acquired in both dry and deluge conditions
in conjunction with time-series aerial photographs would be essential for studying the fill-and-spill mechanism of
prairie wetlands. In this case, we can use the dry-period LiDAR data to delineate and characterize the morphology of
individual wetland depressions before the fill-and-spill processes occur. Furthermore, we can derive the potential
flow paths and project the coalescing of wetland depressions after the fill-and-spill processes initiate. The wet-
period LiDAR data and time-series aerial photographs can serve as validation datasets to evaluate the fill-and-spill
patterns.
It is also worth noting that the proposed methodology in this study was designed to reflect the topography
and hydrologic connectivity between wetlands in the Prairie Pothole Region. We have made assumptions to simplify
the complex prairie hydrology. Physically-based hydrological models (e.g., Brunner and Simmons, 2012; Ameli and
Creed, 2016) have not yet been integrated into our framework. However, fill-and-spill is a complex and spatially
distributed hydrological process highly affected by many factors, such as surface topography, surface roughness, soil
infiltration, soil properties, depression storage, precipitation, evapotranspiration, snowmelt runoff, and groundwater



exchange (Tromp-van Meerveld and McDonnell, 2006a, b; Evenson et al., 2015; Zhao and Wu, 2015; Evenson et
al., 2016; Hayashi et al., 2016). Nevertheless, our study presents the first attempt to use LiDAR data for deriving
nested wetland catchments and simulating flow paths in the broad-scale Pipestem subbasin in the PPR. Previous
studies utilizing high-resolution digital elevation data (e.g., LiDAR, Interferometric Synthetic Aperture Radar
[IfSAR]) for studying prairie wetlands were mostly confined in small-scale areas (e.g., plot scale, small watershed
scale) with a limited number of wetlands, whereas broad-scale studies using physically-based hydrological models
have rarely used LiDAR data to delineate and characterize individual wetland depressions or catchments. Coupled
surface-subsurface flow models with hydrologic, biogeochemical, ecologic, and geographic perspectives have yet to
be developed for broad-scale studies in the PPR (Golden et al., 2014; Amado et al., 2016). Further efforts are still
needed to improve the understanding of the integrated surface-water and groundwater processes of prairie wetlands.
**6 Conclusions**
Accurate delineation and characterization of wetland depressions and catchments are essential for understanding the
hydrology of prairie wetlands. In this study, we accurately delineated the inundation areas while reducing the
confounding factor of live trees by using the LiDAR-derived DEM in conjunction with the coincident LiDAR
intensity imagery. In addition, we developed a semi-automated framework for identifying nested hierarchical
wetland depressions and delineating their corresponding catchments using the localized contour tree method.
Furthermore, we quantified the potential hydrologic connectivity between wetlands and streams based on the
overland flow networks derived using the least-cost path algorithm on LiDAR data. Although the results presented
in this study are specific to the Pipestem subbasin, the proposed framework can be easily adopted and adapted to
other PPR regions, as well as other wetland regions where fine-resolution LiDAR data are available. The new tools
that we developed for identifying hydrologic connectivity between wetlands and stream networks can better inform
wetland regulation debates and enhance the ability to better manage wetlands under various planning scenarios. The
resulting flow network delineated putative temporary or seasonal flow paths connecting wetland depressions to each
other or to the river network at scales finer than available through the National Hydrography Dataset. The results
demonstrated that our proposed framework is promising for improving overland flow modeling and hydrologic
connectivity analysis (Golden et al., 2016).
Broad-scale prairie wetland hydrology has been difficult to study with traditional remote sensing methods.
LiDAR-derived DEMs can be used to map hydrologic flow pathways, which regulate the ability of wetlands to
provide ecosystem services (Lang and McCarty, 2009). This study is an initial step towards the development of a
spatially distributed hydrologic model to fully describe the hydrologic processes in broad-scale prairie wetlands.
Additional field work and the integration of physically-based model of surface and subsurface process would benefit
the study. Importantly, the results capture temporary and ephemeral hydrologic connections and provide essential
information for wetland scientists and decision-makers to more effectively plan for current and future management
of prairie wetlands.




**Data and code availability**

The data and ArcGIS toolbox developed for this paper are available for download at https://GISTools.github.io/.

**Competing interests**

The authors declare that they have no conflict of interest.

**Disclaimer**

This paper has been reviewed in accordance with the U.S. Environmental Protection Agency's peer and administrative review policies and approved for publication. Mention of trade names or commercial products does not constitute endorsement or recommendation for use. Statements in this publication reflect the authors' professional views and opinions and should not be construed to represent any determination or policy of the U.S. Environmental Protection Agency.

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



**Table 1.** Summary statistics of the National Wetlands Inventory (NWI) for the Pipestem subbasin, North Dakota.

| Wetland type | Count | Min (m²) | Max (m²) | Median (m²) | Sum (m²) | Percentage |
|---|---|---|---|---|---|---|
| Freshwater Emergent Wetland | 31,046 | 500 | 3,105,826 | 1,770 | 241,733,542 | 86.5% |
| Freshwater Forested/ Shrub Wetland | 108 | 548 | 343,950 | 2,572 | 1,175,739 | 0.4% |
| Freshwater Pond | 760 | 533 | 719,339 | 1,772 | 14,719,510 | 5.3% |
| Lake | 50 | 3,746 | 9,410,427 | 188,600 | 21,055,438 | 7.5% |
| Riverine | 52 | 634 | 429,838 | 4,021 | 811,488 | 0.3% |
| Total (all polygons) | 32,016 | 500 | 9,410,427 | 1,778 | 279,495,717 | 100.0% |






**Table 2.** Summary statistics of NWI wetland polygons and inundation polygons derived from LiDAR intensity data.

| Type | Count | Min (m$^2$) | Max (m$^2$) | Mean (m$^2$) | Median (m$^2$) | Sum (m$^2$) |
|------|-------|---------|---------|----------|------------|---------|
| NWI polygons | 32,016 | 500 | 9,410,427 | 8,728 | 1,778 | 279,495,717 |
| Inundation polygons | 15,784 | 500 | 7,348,000 | 17,650 | 1,825 | 278,523,863 |
| Dried NWI polygons | 18,957 | 500 | 112,100 | 2,299 | 1,212 | 43,574,627 |






**Table 3**. Summary statistics of 33,241 wetland depressions and catchments derived from LiDAR DEM.

| Type | Min | Max | Mean | Median | Sum |
|---|---|---|---|---|---|
| Depression area (m$^2$) | 1008 | 20,030,000 | 16,590 | 2592 | 554,506,299 |
| Catchment area (m$^2$) | 1818 | 57,900,000 | 82,710 | 25,780 | 2,770,116,549 |
| Depression volume (m$^3$) | 1 | 153,000,000 | 23,420 | 420 | 782,886,383 |
| Proportion of depression area to catchment area (%) | 0.04 | 83.72 | 16.59 | 14.31 | 20.06 |






**Table 4**. Summary statistics of wetland depression ponding depth, NHD flowlines, connectivity length, and
elevation difference.

| Type | Count | Min (m) | Max (m) | Mean (m) | Median (m) | Sum (m) |
|---|---|---|---|---|---|---|
| Ponding depth | 33,241 | 0.01 | 7.64 | 0.23 | 0.16 | NA |
| NHD flowlines | 1840 | 3.89 | 15,530 | 762 | 317 | 1,402,226 |
| Connectivity length | 41,449 | 1.5 | 4,658 | 138 | 83 | 5,014,495 |
| Elevation difference | 41,449 | 0.01 | 70.89 | 2.14 | 0.89 | NA |






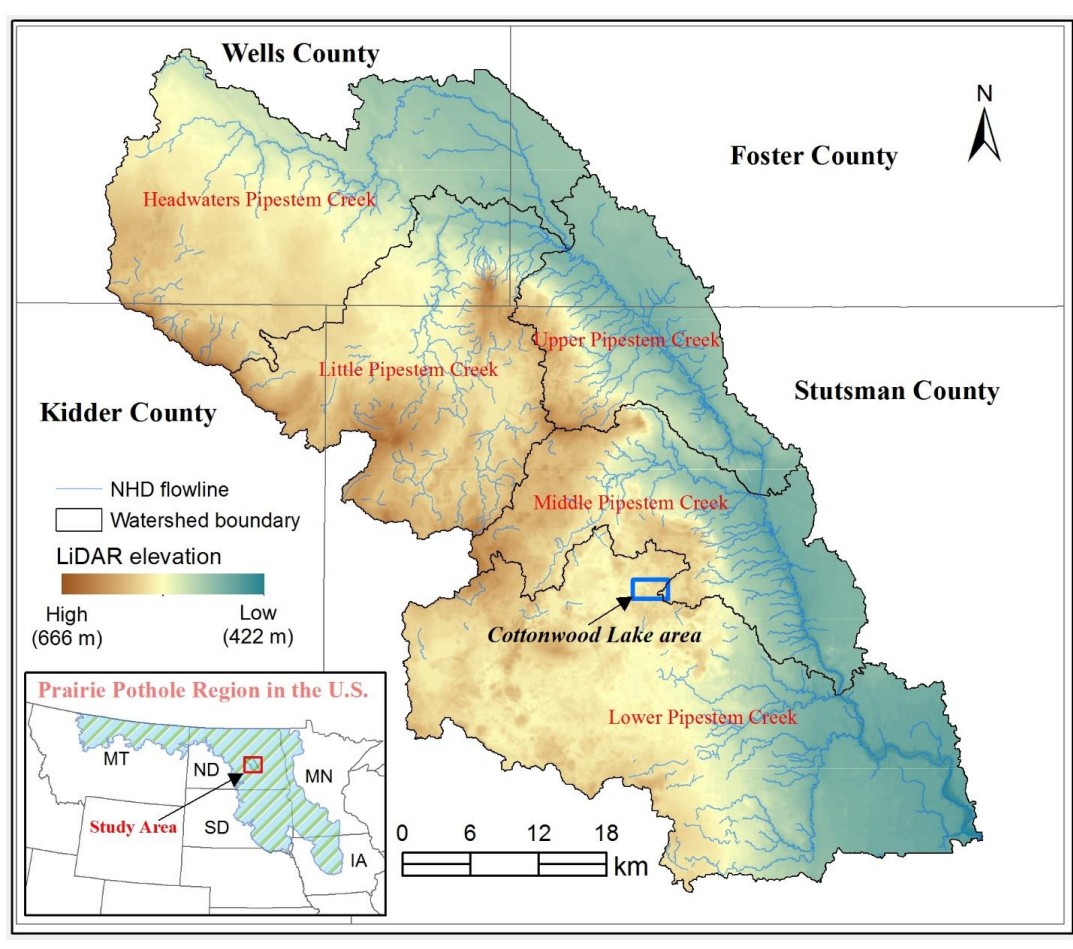


**Figure 1.** Location of the Pipestem subbasin within the Prairie Pothole Region of North Dakota.





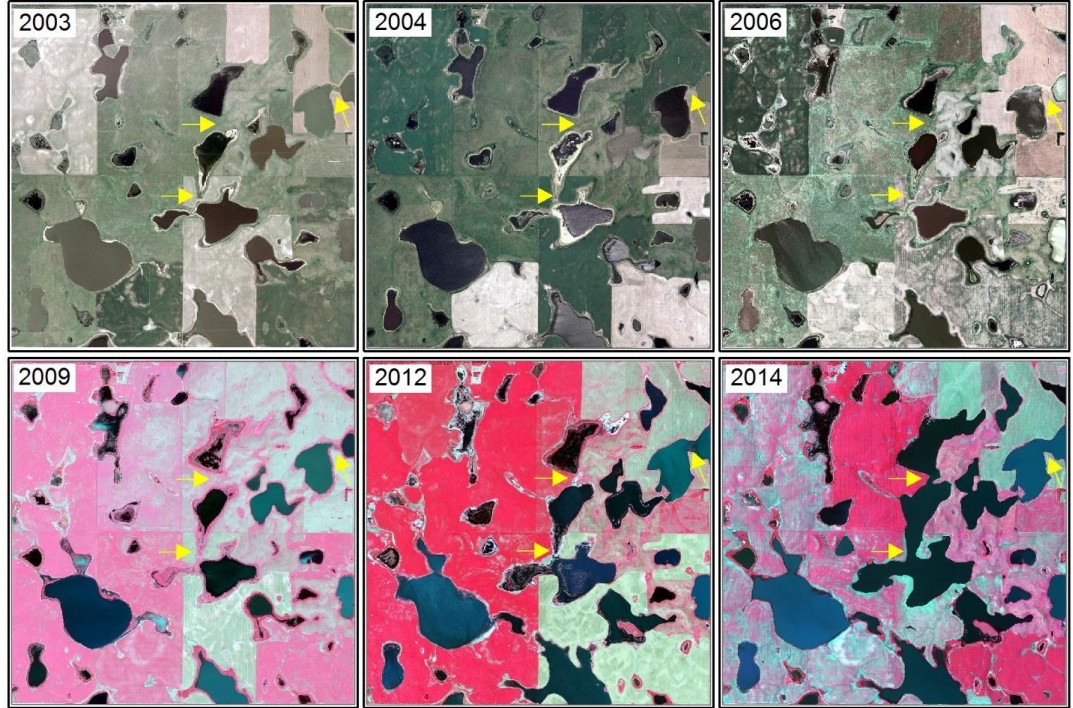


**Figure 2.** Examples of the National Agriculture Imagery Program (NAIP) aerial imagery in the Prairie Pothole
Region of North Dakota illustrate the dynamic nature of prairie pothole wetlands under various dry and wet
conditions. The yellow arrows highlight locations where filling-spilling-merging dynamics occurred (imagery
location: 99°8'34.454" W, 47°1'23.519" N).





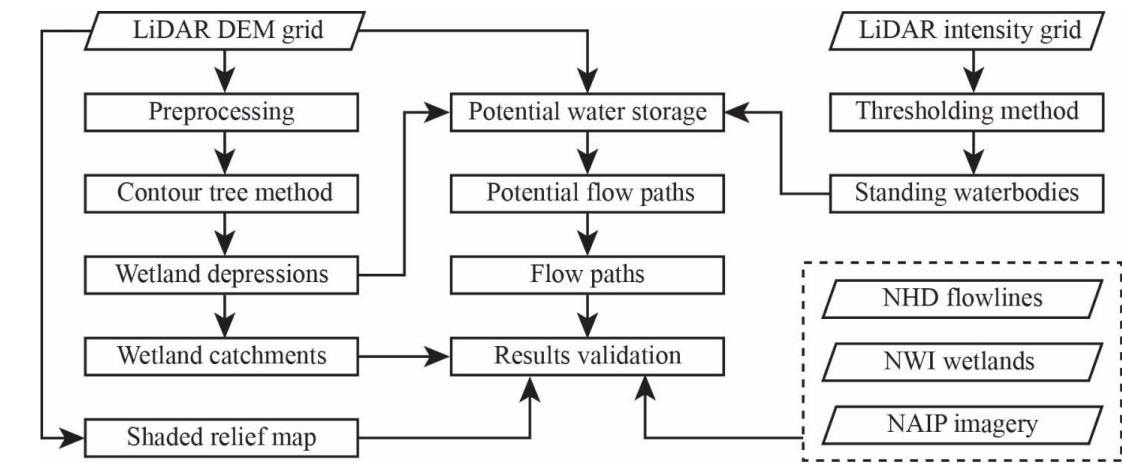


**Figure 3**. Flowchart of the methodology for delineating wetland catchments and flow paths.





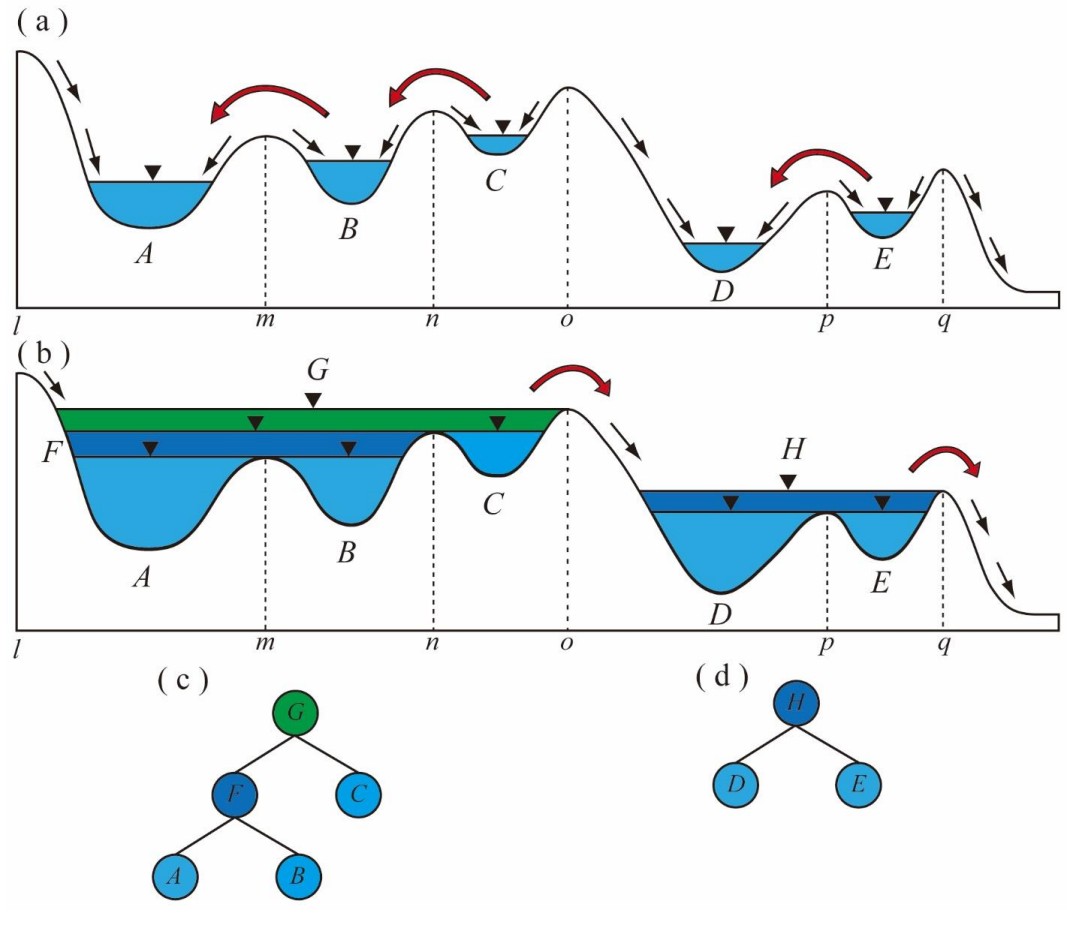


**Figure 4.** Illustration of the filling-merging-spilling dynamics of wetland depressions: (a) first-level depressions; (b) nested hierarchical structure of depressions under fully-filled condition; (c) corresponding contour tree representation of the composite wetland depression (left) in (a); and (d) corresponding contour tree representation of the composite wetland depression (right) in (a). Different color of nodes in the tree represents different portions of the composite depression in (a): light blue (first-level), dark blue (second-level), and green (third-level).





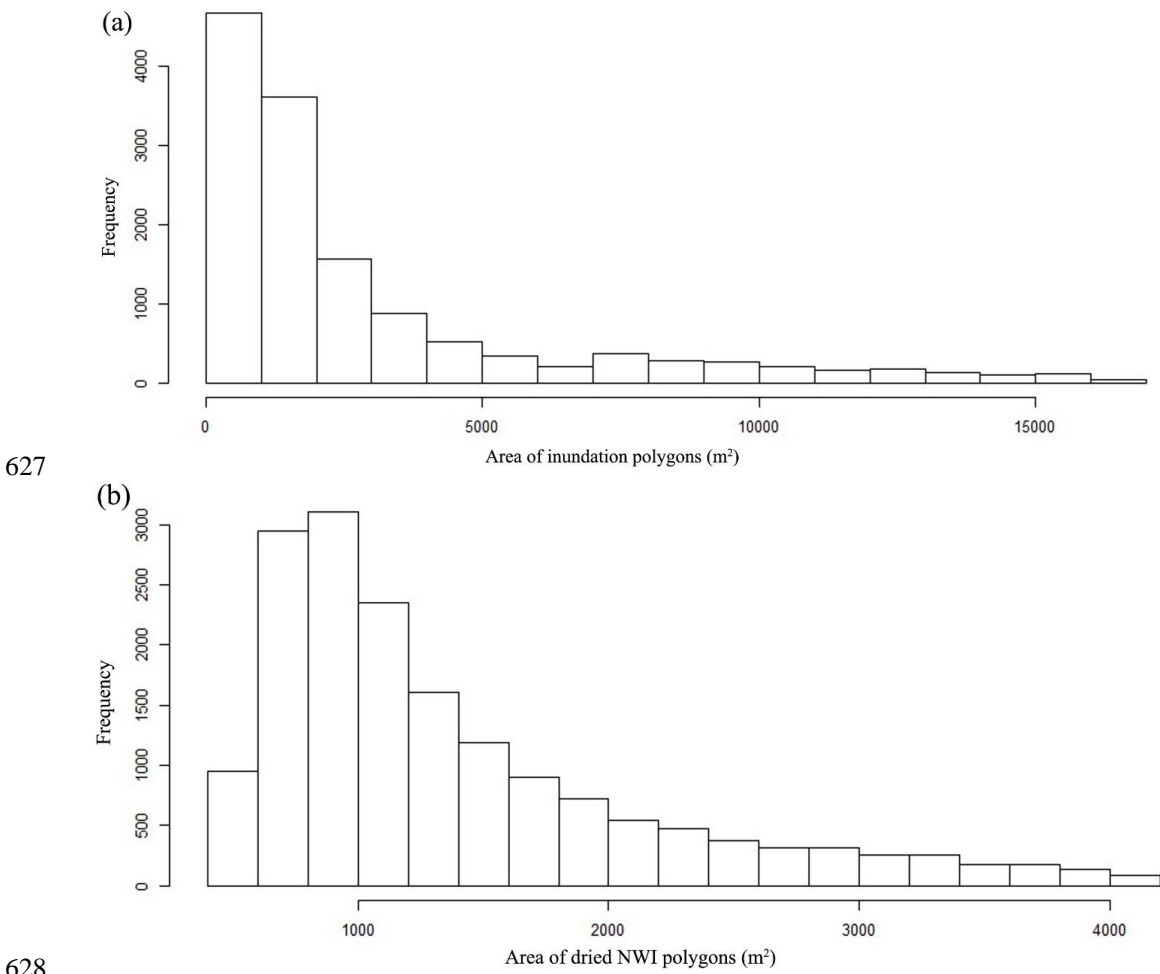



**Figure 5.** Histograms of inundation and NWI wetland polygons. (a) Inundation objects derived from LiDAR
intensity data; (b) dried NWI wetland polygons not intersecting inundation objects.







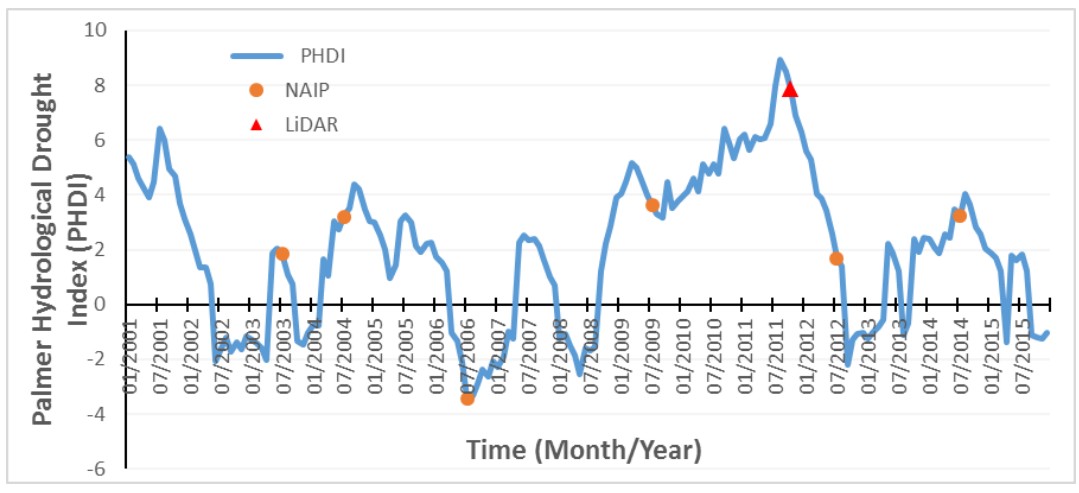


**Figure 6.** Palmer Hydrological Drought Index (PHDI) of the Pipestem subbasin (2001-2015).





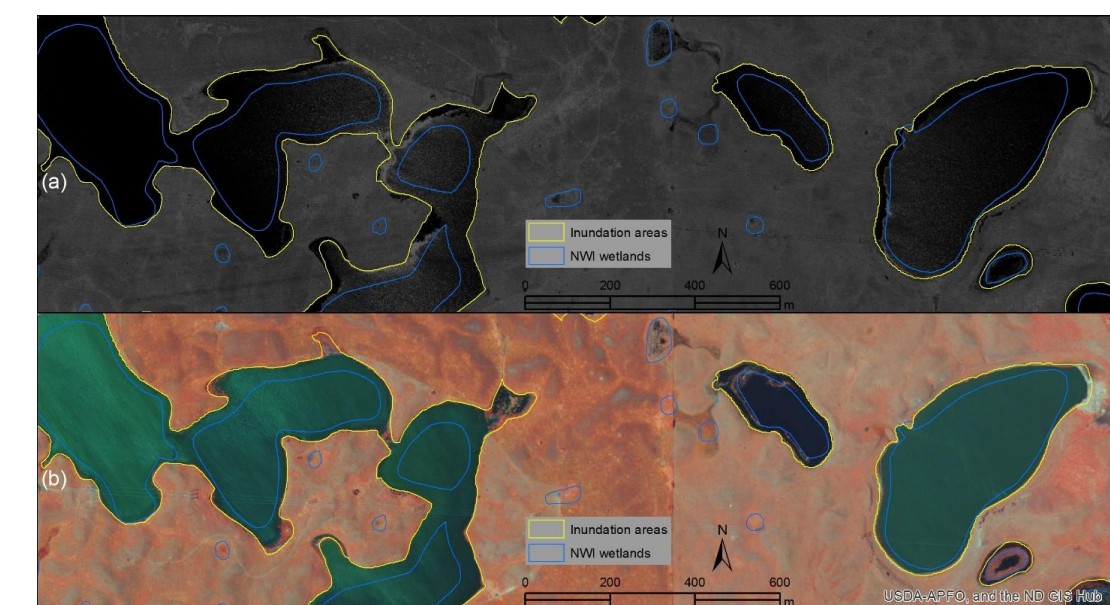

**Figure 7.** Comparison between inundation areas (derived from LiDAR intensity data) and NWI wetland polygons (image location: 99°9'53.9" W, 47°3'34.474" N). (a) Inundation areas and NWI wetlands overlaid on LiDAR intensity image; and (b) inundation areas and NWI wetlands overlaid on color infrared aerial photograph (2009).



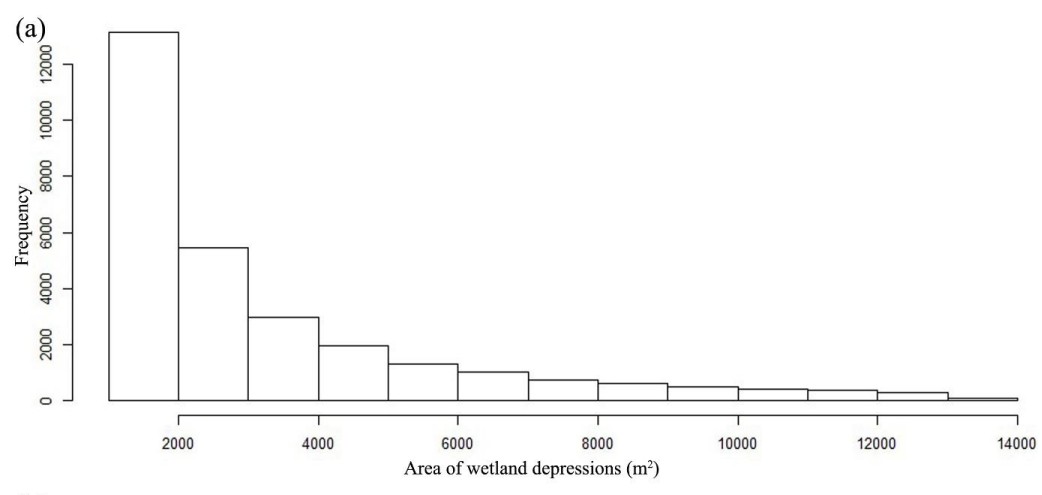


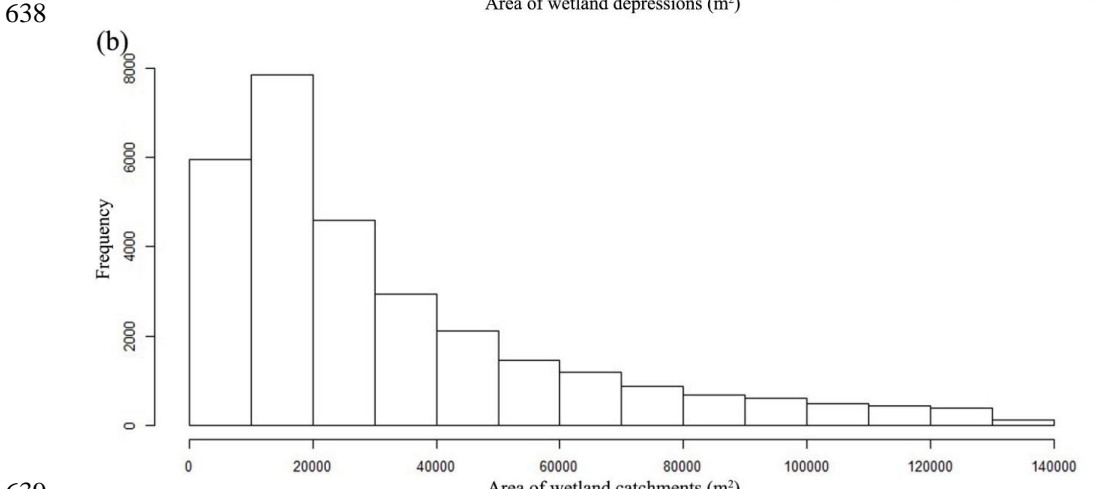


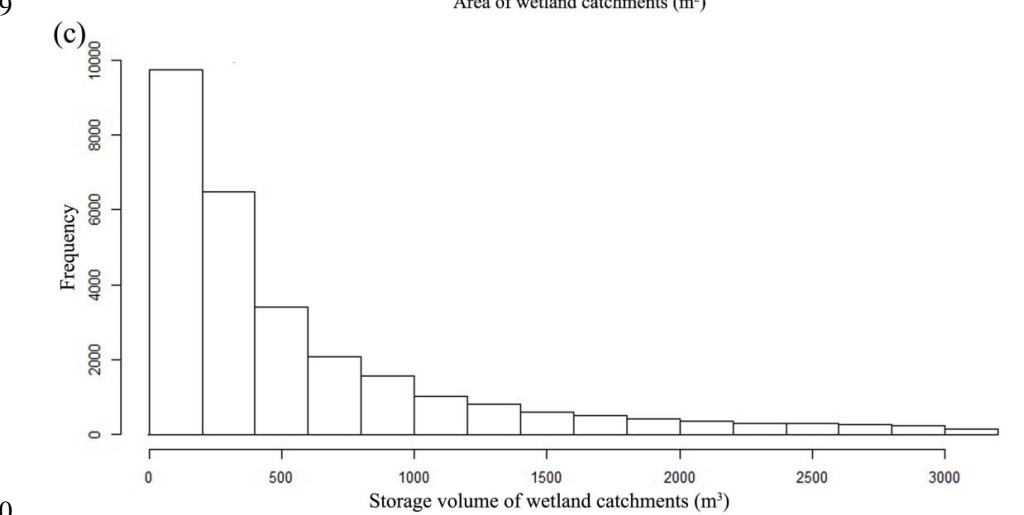






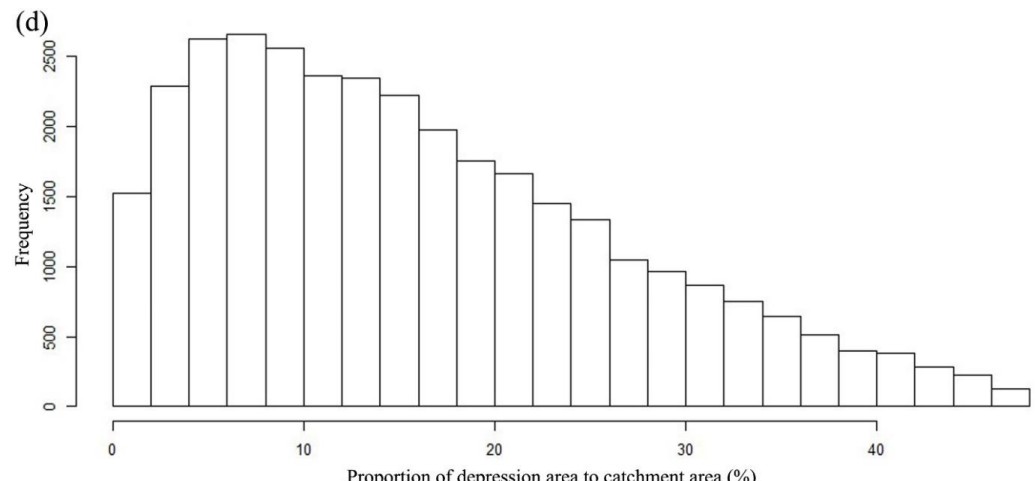

**Figure 8.** Histogram of wetland depressions and catchments. (a) Wetland depressions; (b) wetland catchments; (c)
potential storage capacity; and (d) proportion of depression area to catchment area.





**Figure 9**. Examples of LiDAR-derived wetland depressions and flow paths in the Pipestem subbasin (image location: 98°59'48.82" W, 47°1'32.679" N). (a) Wetland depressions and flow paths overlaid on LiDAR shaded relief map; and (b) NWI polygons, wetland depressions and flow paths overlaid on color infrared aerial photograph (2012).









**Figure 10.** Histogram of wetland connectivity. (a) Connectivity lengths; and (b) elevation differences between connected wetlands.

