# Peer review of "Delineating wetland catchments and modeling hydrologic connectivity using LiDAR data and aerial imagery"

_Hydrology and Earth System Sciences, 2017_

## Referee Comment (RC1) · Anonymous Referee #1 · 25 Feb 2017

Manuscript: Delineating wetland catchments and modeling hydrologic connectivity using LiDAR data and aerial imagery

General Comments

This manuscript addresses an important area of research, improving our ability to model and map hydrological interactions between wetlands and streams. It is well written and creative in its integration of methods, however the paper becomes a little confusing and muddled in interpreting whether theoretical or actual connectivity was modeled. In addition the inundation map does not appear to be validated. General and specific comments are below.

1) The Introduction provides a strong background of the PPR, but could be strengthened by clarifying the novelty of the approach. Right now this is limited to stating that few studies of prairie wetlands have treated wetlands and catchments as integrated units and lidar is rarely used at broad scale but no citations are offered. Has this approach been used in other wetland landscapes, just not in the PPR? Or is this approach actually quite novel? What about related studies that have mapped wetland depressions and/or delineated wetland catchments? How does this approach fit in with those studies? Adding just a few sentences to discuss this would help contextualize the work presented.

2) Right now the last paragraph of the introduction is essentially a summary of what the paper did, but it would be stronger if the authors added what the goals, objectives, or research questions were. . .for example, our goal was to demonstrate a method to map potential hydrologic connections between wetlands and the river networks. . .

3) In calculating flow paths it is sometimes acknowledged that these are potential and sometimes stated that temporary and intermittent flowpaths have been identified. It should be made clear that these are potential hydrologic connections that are identified via the flowpaths, as it is not shown currently in the paper how or if active flowpaths are or could be distinguished from inactive flowpaths. However, the authors also mapped inundation and depressions, couldn't these be used to determine which depressions were connected? It isn't entirely clear if this is what is presented partially in Figure 10 or not.

4) It does not appear that the inundation map has been validated. Because of a lack of date match between the NAIP imagery and the LiDAR collection date, the NAIP imagery, appropriately, is primarily used to show that surface water changes over time. I realize it is challenging to validate maps classified from high resolution imagery but given the nearby Cottonwood site which monitors water levels at multiple ponds, are there field-measured water levels collected at a close date that could be used to help validate the inundation map?

5) I think the conclusion that dry NWI wetlands are likely no longer wetlands is a bit of a stretch. The PHDI does not consider snowmelt and is just based on rainfall and temperature, as the LiDAR collection was in October it is entirely possible that a large number of these wetlands are temporarily wet for few weeks in the spring following snowmelt. I don't think you can assume that these NWI wetlands no longer function as wetlands given just 1 fall date of inundation, even if in a wet year.

Minor Comments Line 27 – grammatical error, change "highly" to "most" and modify sentence to avoid using "as" twice.

Line 32 – awkward sentence, change to "the potholes range in size from"

Line 34 – the term ephemeral is more commonly used for streams, the term "temporary" is more commonly used for wetlands.

Line 37 – remove the word "as"

Line 39 – conterminous is misspelled

Line 94 – change to "collected in late October"

Line 97 – add space between in and 15.0 cm.

2.2. LiDAR Data – I realize you mention this in the Discussion, but it would be helpful to also add quick comment here regarding how wet October 2011 was and how this may have influenced the resulting DEM.

Line 126 – change "these" to "the"

Comment - In the Methodology section change from present tense to past tense.

Line 215 – add the word "of" between number and upslope.

Section 3.2 – I'm assuming to use the contour approach you need to convert the DEM to vectors. . .is any information lost in this process?. . .why not just use a raster-based approach to find depressions?

Section 3.4 – In calculating ponding time, are you assuming no infiltration? If so, add this as an assumption to the text.

Section 3.4 – Does the water storage capacity, and in turn the ponding time equations assume the depression is dry to start with? How is the pre-existing water in the depressions dealt with? This is particularly an issue for permanent wetlands.

Comment – what was the range of rainfall intensities that were added to derive the ponding time estimates?

Line 287-310 This paragraph is methods and should be moved to the Methods section accordingly.

Line 303-304 – What about inundation in streams that may not have been mapped as depressions, would these inundation objects be lost given this filtering step?

Section 4.2 How common was it for wetland depressions to be nested within a larger catchments? Is there a way this nested hierarchy could be quantified or showed?

Line 362-363 – Although the findings are based on a much larger sample size, they are also all derived from a single watershed, so the results may also be site specific.

Section 4.3 – The flow paths are potential flow paths, however, right? Water may not have flowed along a fraction of them to date. This should be made clear in the text.

Line 384 – remove "the" before late October

Line 385 – add "a" after such.

Line 388-389 - revise sentence to "A substantial number of inundated NWI wetlands were found to coalesce with adjacent wetlands. . .."

Line 402 – Do you mean you "could" use it if a dry-period LiDAR was available?

Line 384-406 – This is a good discussion of an important issue but it is not entirely clear how this issue affected your findings in this case. I would guess that you likely

under-estimated the number of depressions that coalesced?

Line 404-405 – As far as I can tell, however, in this case you did not use the time-series or wet inundation to evaluate or summarize fill-and-spill patterns. Is this correct?

Line 425 – Can't use the word "accurately" if no validation was done.

Line 433 – Add "potential" before hydrological connectivity.

Line 435 – I am struggling with this statement which is used several times throughout the manuscript. Although temporary or seasonal flow paths were likely identified, flow-paths were also likely identified that never actually carry water. How can we distinguish between these or can we?

Line 439 –Add what the specific limiting factors have been with traditional remote sensing methods.

Table 1 – remove extra spaces between Freshwater and Emergent.

Figure 5, 8 and 10 – I would add a basic color to the histograms, maybe light gray? To improve the aesthetics.

Figure 6 – Modify x-axis to just show year

Figure 7 – the yellow and blue lines are hard to see, maybe making them a little thicker might make them more visible.

Figure 10 – This figure gets at several questions I had. Was connectivity calculated so that all wetlands connected to each other and eventually to a stream? And this is then the length distribution of those flowpath lines? If so it should be indicated that these are potential connectivity. What does connected wetlands mean here? Are these just the coalesced wetlands?

---

## Referee Comment (RC2) · Anonymous Referee #2 · 17 Mar 2017

This manuscript was well thought out, well organized and well written. In the United States the regulatory status of wetlands is currently linked to connectivity to streams so the topic of this manuscript is important. The conceptual model presented for wetland fill and spill seems very useful. The approach used in the reported study is sound and findings support the conclusions reached.

Specific comments:

The last paragraph of the introduction is a summary of the study findings. It should be modified to reflect study goals instead.

Flow routing was performed using D8 algorithm (line 213) but often it has been found

that D-infinity algorithms provide more realistic flow characteristics.

When reporting numerical results consider the errors associated with the underlying model used to produce the values. The number of nonzero digits should generally reflect the uncertainty. For example see lines 347 and 348 with values reported with 4 significant figures whereas it is known that these estimates have substantial uncertainty. Also in tables with data reported with up to 8 significant digits (Tables 1 to 4).

Figure 7 needs to be reworked. Labels on figure are very difficult to read

---

## Author Comment (AC2) · 22 Mar 2017

General Comments This manuscript was well thought out, well organized and well written. In the United States the regulatory status of wetlands is currently linked to connectivity to streams so the topic of this manuscript is important. The conceptual model presented for wetland fill and spill seems very useful. The approach used in the reported study is sound and findings support the conclusions reached.

RESPONSE: We thank the reviewer for the encouraging comments.

Specific comments:

The last paragraph of the introduction is a summary of the study findings. It should be

modified to reflect study goals instead.

RESPONSE: We thank the reviewer for the good suggestion. We will revise this paragraph and make our research objectives more clear.

Flow routing was performed using D8 algorithm (line 213) but often it has been found that D-infinity algorithms provide more realistic flow characteristics.

RESPONSE: We agree with the reviewer that D-infinity algorithms might provide more realistic flow characteristics. In our study, the flow direction raster was generated and used as an intermediate dataset to derive wetland catchments. For delineating catchments/watersheds, we tried the ArcGIS Hydrology Toolbox (https://goo.gl/GhmFId) and the open-source Whitebox Geospatial Analysis Tools (https://goo.gl/dqV4cE). Both software packages use D8 algorithm for watershed delineation. Since our data processing flow was built on the ArcGIS Hydrology Toolbox, for the sake of simplicity, we used the D8 algorithm available in ArcGIS to derive flow directions. Nevertheless, we believe that both flow direction algorithms should lead to the same watershed delineation results.

When reporting numerical results consider the errors associated with the underlying model used to produce the values. The number of nonzero digits should generally reflect the uncertainty. For example see lines 347 and 348 with values reported with 4 significant figures whereas it is known that these estimates have substantial uncertainty. Also in tables with data reported with up to 8 significant digits (Tables 1 to 4).

RESPONSE: We appreciate this concern. In the revised manuscript, we will reduce the number of significant digits to no more than two throughout the manuscript.

Figure 7 needs to be reworked. Labels on figure are very difficult to read

RESPONSE: We have revised Figure 7. We made the lines thicker. In addition, we switched the line colors to make them consistent with those shown in Figure 9. Yellow line and blue line represent NWI wetlands and LiDAR-derived inundation areas,

respectively.

Please also note the supplement to this comment:
http://www.hydrol-earth-syst-sci-discuss.net/hess-2017-1/hess-2017-1-AC2-supplement.pdf

[Figure]

[Figure]

**Fig. 1.** Figure 7. Comparison between inundation areas (derived from LiDAR intensity data) and NWI wetland polygons

---

## Author Response (AR1)

**Our Response to Anonymous Referee #1**

**General Comments**

This manuscript addresses an important area of research, improving our ability to model and map hydrological interactions between wetlands and streams. It is well written and creative in its integration of methods, however the paper becomes a little confusing and muddled in interpreting whether theoretical or actual connectivity was modeled. In addition the inundation map does not appear to be validated. General and specific comments are below.

**RESPONSE**: We thank the reviewer for his/her thorough review and very helpful comments/suggestions. The positive feedback encourages us to continue working on this subject in the future.

1) The Introduction provides a strong background of the PPR, but could be strengthened by clarifying the novelty of the approach. Right now this is limited to stating that few studies of prairie wetlands have treated wetlands and catchments as integrated units and lidar is rarely used at broad scale but no citations are offered. Has this approach been used in other wetland landscapes, just not in the PPR? Or is this approach actually quite novel? What about related studies that have mapped wetland depressions and/or delineated wetland catchments? How does this approach fit in with those studies? Adding just a few sentences to discuss this would help contextualize the work presented.

**RESPONSE**: We thank the reviewer for the useful comments. We have revised this section and added appropriate references to justify the novelty of our approach.

 "To our knowledge, little work has been done to delineate potential flow paths between wetlands and stream networks and use flow paths to characterize hydrologic connectivity in the PPR. In addition, previous remote sensing-based work on the hydrology of prairie wetlands mainly focused on mapping wetland inundation areas (e.g., Huang et al., 2014; Vanderhoof et al., 2017) or wetland depressions (e.g., McCauley and Anteau, 2014; Wu and Lane, 2016), few studies have treated wetlands and catchments as integrated hydrological units. Therefore, there is a call for treating prairie wetlands and catchments as highly integrated hydrological units because the existence of prairie wetlands depends on lateral inputs of runoff water from their catchments in addition to direct precipitation (Hayashi et al., 2016). Furthermore, hydrologic models for the PPR were commonly developed using coarse-resolution DEMs, such as the 30-m National Elevation Dataset (see Chu, 2015; Evenson et al., 2015; Evenson et al., 2016). High-resolution light detection and ranging (LiDAR) data have rarely been used in broad-scale (e.g., basin- or subbasin-scale) studies to delineate wetland catchments and model wetland connectivity in the PPR."

2) Right now the last paragraph of the introduction is essentially a summary of what the paper did, but it would be stronger if the authors added what the goals, objectives, or research questions were. . .for example, our goal was to demonstrate a method to map potential hydrologic connections between wetlands and the river networks. . .

**RESPONSE**: Good suggestion. We have revised this paragraph and made our research objectives clear.

"In this paper, we present a semi-automated framework for delineating nested hierarchical wetland depressions and their corresponding catchments as well as simulating wetland connectivity using high-resolution LiDAR data. Our goal was to demonstrate a method to characterize fill-spill wetland hydrology and map potential hydrological connections between wetlands and stream networks. The hierarchical structure of wetland depressions and catchments was identified and quantified using a localized contour tree method (Wu et al., 2015). The potential hydrologic connectivity between wetlands and streams was characterized using the least-cost path algorithm. We also utilized high-resolution LiDAR intensity data to delineate wetland inundation areas, which were compared against the National Wetlands Inventory (NWI) to demonstrate the hydrological dynamics of prairie wetlands. Our ultimate goal is to build on our proposed framework to improve overland flow simulation and hydrologic connectivity analysis, which subsequently may improve the understanding of wetland hydrological dynamics at watershed scales."

3) In calculating flow paths it is sometimes acknowledged that these are potential and sometimes stated that temporary and intermittent flowpaths have been identified. It should be made clear that these are potential hydrologic connections that are identified via the flowpaths, as it is not shown currently in the paper how or if active flowpaths are or could be distinguished from inactive flowpaths. However, the authors also mapped inundation and depressions, couldn't these be used to determine which depressions were connected? It isn't entirely clear if this is what is presented partially in Figure 10 or not.

**RESPONSE**: In the revised version, we have made it clear that the derived flow paths are potential hydrologic connections. We tried to distinguish active flow paths from inactive flow paths by visually assessing the derived flow paths overlaid on aerial photographs. Although we were not able to conduct quantitative assessments, we did find that the many potential flow paths were collocated with vegetated areas, which indicates that flow paths are likely located in high soil moisture areas that are directly or indirectly related to surface water or groundwater connectivity. The inundation areas were derived from the LiDAR intensity data, whereas the depressions were derived from the LiDAR DEM. Theoretically, the inundation areas are a subset of the depressions. In other words, the depressions represent the maximum ponded extents for the inundation areas. Since the LiDAR data were acquired during an extremely wet period, many wetlands already coalesced with adjacent wetlands to form larger wetland complexes. We could use dry-period aerial photographs to determine which depressions were connected. The LiDAR data can be used to show which depressions might potentially connect, but it cannot tell which depressions were actually connected.

4) It does not appear that the inundation map has been validated. Because of a lack of date match between the NAIP imagery and the LiDAR collection date, the NAIP imagery, appropriately, is primarily used to show that surface water changes over time. I realize it is challenging to validate maps classified from high resolution imagery but given the nearby Cottonwood site which monitors water levels at multiple ponds, are there field-measured water levels collected at a close date that could be used to help validate the inundation map?

**RESPONSE**: We thank the reviewer for this good suggestion. We have looked into the in-situ water-level data of the Cottonwood Lake Study Area retrieved from the USGS website (Mushet et al. 2016). We chose the field-measured water levels collected on October 27, 2011, which was the closest date to the LiDAR data acquisition date used in this study. The water levels of eight semi-permanent wetlands numbered through P1-P8 were used to compare the water elevation of wetland inundation areas derived using LiDAR data. The water level difference between the field measurement and LiDAR-derived measurement for these eight wetlands ranged from -5 cm to 11 cm, with an average elevation difference of 0.5 cm, which falls within the vertical accuracy of the LiDAR DEM (15.0 cm). We have also made the inundation map available online at http://wetlands.io/maps/inundation.html.

| Wetland ID | P01 | P02 | P03 | P04 | P06 | P07 | P08 | Average |
|---|---|---|---|---|---|---|---|---|
| Field-measured (m) | 560.30 | 561.01 | 557.86 | 561.01 | 562.55 | 562.76 | 556.66 | NA |
| LiDAR-derived (m) | 560.22 | 561.12 | 557.91 | 561.12 | 562.50 | 562.71 | 556.61 | NA |
| Water-level difference (m) | -0.08 | 0.11 | 0.05 | 0.11 | -0.05 | -0.05 | -0.05 | 0.0057 |

David M. Mushet, Donald O. Rosenberry, Ned H. Euliss, Jr., and Matthew J. Solensky, 2016, Cottonwood Lake Study Area - Water Surface Elevations. U.S. Geological Survey Data Release, http://dx.doi.org/10.5066/F7707ZJ6

5) I think the conclusion that dry NWI wetlands are likely no longer wetlands is a bit of a stretch. The PHDI does not consider snowmelt and is just based on rainfall and temperature, as the LiDAR collection was in October it is entirely possible that a large number of these wetlands are temporarily wet for few weeks in the spring following snowmelt. I don't think you can assume that these NWI wetlands no longer function as wetlands given just 1 fall date of inundation, even if in a wet year.

**RESPONSE**: We thank the reviewer for this concern. In the revised version, we have removed the statement that those 'dry' NWI wetlands are no longer wetlands. We have added more explanations about the discrepancy between NWI wetlands and our results derived by the LiDAR data.

"It is worth noting that most of these 'dried' NWI wetlands were relatively small with a median size of $1.2 \times 10^3$ m$^2$ (Table 2). The LiDAR intensity data were acquired in late October 2011, an extremely wet month according to the Palmer Hydrological Drought Index (Fig. 6). During this wet season, most wetlands would be expected to have abundant standing water. If no standing water could be detected in a wetland patch during this extremely wet period, it is possible that some of these small wetlands might have dried out during the past weeks to months. It is possible that land use change surrounding the 'dried' wetlands (e.g., row-cropping replacing pasture lands) may have affected their hydrology (Wright and Wimberly, 2013); water diversion via drainage or ditches could also be responsible for the lack of inundation, though we did not explore either of these potential drivers of change in this study. However, it is also likely that some of the 'dried' wetland might become wet again in the spring following snowmelt. The 'dried' NWI wetlands could also be attributed to the source of error in the original NWI data, which has a minimum mapping unit (i.e., the minimum sized wetland that can be consistently mapped) of 0.1 ha for the PPR (Tiner, 1997). Figure 5(b) shows that 37% of the 'dried' NWI polygons are smaller than the minimum mapping unit (1000 m$^2$). This implies that these small 'dried' NWI polygons could be due to the NWI mapping error."

**Minor Comments**

Line 27 – grammatical error, change "highly" to "most" and modify sentence to avoid using "as" twice.

**RESPONSE**: We have revised the sentence as suggested.

Line 32 – awkward sentence, change to "the potholes range in size from"

**RESPONSE**: We have revised the sentence as suggested.

Line 34 – the term ephemeral is more commonly used for streams, the term "temporary" is more commonly used for wetlands.

**RESPONSE**: We have changed "ephemeral" to "temporary"

Line 37 – remove the word "as"

**RESPONSE**: We have removed "as".

Line 39 – conterminous is misspelled

**RESPONSE**: We have corrected the typo.

Line 94 – change to "collected in late October"

**RESPONSE**: We have revised the sentence as suggested.

Line 97 – add space between in and 15.0 cm.

**RESPONSE**: Space added.

2.2. LiDAR Data – I realize you mention this in the Discussion, but it would be helpful to also add quick comment here regarding how wet October 2011 was and how this may have influenced the resulting DEM.

**RESPONSE**: We thank the author for the good suggestion. We have added the following sentences to the end of this section: "It is worth noting that October 2011 was an extreme wet period according to the Palmer Hydrological Drought Index. Consequently, small individual wetland depressions nested within larger inundated wetland complexes might not be detectable from the resulting LiDAR DEM."

Line 126 – change "these" to "the"

**RESPONSE**: We have changed "these" to "the".

Comment - In the Methodology section change from present tense to past tense.

**RESPONSE**: We have revised the Methodology section as suggested. For specific data processing steps we performed, we used the past tense. When describing diagrams not tied to specific data processing steps we performed, we used the present tense.

Line 215 – add the word "of" between number and upslope.

**RESPONSE**: We have added the word "of".

Section 3.2 – I'm assuming to use the contour approach you need to convert the DEM to vectors. . .is any information lost in this process? why not just use a raster-based approach to find depressions?

**RESPONSE**: To the best of our knowledge, there is no raster-based approach available to delineate and characterize the nested hierarchical structure of depressions. The traditional sink-filling method can delineate the maximum extent of a composite depression. However, it cannot distinguish or delineate the individual depressions (if any) nested within the composite depression. Moreover, the topological relationship between nested depressions cannot be derived from the raster-based approach. The vector-based contour-tree approach used in this study can not only identify nested depressions but also characterize their topological relationships, which are crucial for studying the filling-merging-spilling hydrology. In this study, we set the contour interval as 20 cm, which was chosen based on the LiDAR vertical accuracy (15 cm) and consideration of computational time. Like any other vector-raster data conversion process, there might be some information lost in this process. Nevertheless, we believe that the contour interval of 20 cm is sufficient to minimize the information loss based on our experiments. An in-depth discussion about the contour interval selection is available in the Wu et al. (2015) paper.

Section 3.4 – In calculating ponding time, are you assuming no infiltration? If so, add this as an assumption to the text.

**RESPONSE**: We have added the following sentences to this section: "For the sake of simplicity, we made two assumptions. First, we assumed that the rainfall was temporally and spatially consistent and uniformly distributed throughout the landscape and all surfaces were impervious. Second, we assumed no soil infiltration."

Section 3.4 – Does the water storage capacity, and in turn the ponding time equations assume the depression is dry to start with? How is the pre-existing water in the depressions dealt with? This is particularly an issue for permanent wetlands.

**RESPONSE**: We did not assume the depression is dry to start with. The ponding time of a depression was calculated based on the potential water storage and its corresponding catchment area. In other words, we don't need the existing water storage to calculate the ponding time. It does not matter whether a depression is dry or has existing water. If a depression is completely dry without any existing water, the potential water storage refers to the storage volume between the lowest point and the spilling point of the depression. If a depression has existing water in it, the potential water storage refers to the potential water volume the depression can hold between the water surface and the spilling point. The potential water storage capacity of each wetland depression was computed through statistical analysis of the LiDAR DEM grid cells that fall within the depression. The calculation of existing water storage is not the focus of this study. Since the near-infrared LiDAR sensors generally could not penetrate water, the depression morphology beneath the water surface could not be derived from LiDAR data. Therefore, it is not possible to calculate the exact storage volume of an existing waterbody. However, many studies have showed that there is a strong statistical relationship between storage volume and surface area in a depression (e.g., see Gleason, et al. (2007), Wu and Lane (2016)). Therefore, existing water storage can be estimated using empirical equations if needed.

1) Gleason, R. A., B. A. Tangen, M. K. Laubhan, K. E. Kermes and N. H. Euliss Jr. 2007. Estimating water storage capacity of existing and potentially restorable wetland depressions in a subbasin of the Red River of the North. p. 36. U.S. Geological Survey Open-File Report 2007-1159.
2) Wu, Q. and C. R. Lane. 2016. Delineation and Quantification of Wetland Depressions in the Prairie Pothole Region of North Dakota. Wetlands, 36:215-227.

Comment – what was the range of rainfall intensities that were added to derive the ponding time estimates?

**RESPONSE**: For this study, we used a uniform steady rainfall with an intensity of 5.0 cm/h based on the literature (e.g., see Chu (2015)). The rainfall intensity can be easily adapted in other study areas to derive the ponding time estimates.

1) Chu, X. 2015. Delineation of Pothole-Dominated Wetlands and Modeling of Their Threshold Behaviors. Journal of Hydrologic Engineering: D5015003.

Line 287-310 This paragraph is methods and should be moved to the Methods section accordingly.

**RESPONSE**: We have moved this paragraph to the Methodology section as suggested.

Line 303-304 – What about inundation in streams that may not have been mapped as depressions, would these inundation objects be lost given this filtering step?

**RESPONSE**: Based on our visual assessments, no major inundation areas along streams were missed during the filtering step. In other others, all major inundation areas along streams were mapped as depressions.  The inundation mapping results overlaid on the LiDAR intensity imagery are available for viewing at http://wetlands.io/maps/inundation.html.

Section 4.2 How common was it for wetland depressions to be nested within a larger catchments? Is there a way this nested hierarchy could be quantified or showed?

**RESPONSE**: The fill-and-spill hydrology in the Prairie Pothole Region is well documented in the literature. It is very common for wetland depression to be nested within a larger catchment. In our study, the nested hierarchy was quantified and characterized using the localized contour tree approach. A conceptual diagram of the approach is shown in Figure 4. Real-world examples demonstrating the fill-spill process and nested hierarchy can be seen from the time-series aerial images shown in Figure 2.

Line 362-363 – Although the findings are based on a much larger sample size, they are also all derived from a single watershed, so the results may also be site specific.

**RESPONSE**: It is true that our results are also site-specific. We have modified the sentence and stated that the results are "for the study area" only. Nevertheless, we believe that our results regarding the proportion of depression area to catchment area are statistically more reliable than that reported in previous studies, which were calculated based on a very limited number (<20) of depressions. In contrast, our results were computed from more than 30,000 wetland depressions and catchments.

Section 4.3 – The flow paths are potential flow paths, however, right? Water may not have flowed along a fraction of them to date. This should be made clear in the text.

**RESPONSE**: We thank the reviewer for pointing this out. We have modified the text accordingly and made it clear that the flow paths derived in this study are potential flow paths.

Line 384 – remove "the" before late October

**RESPONSE**: We have removed "the".

Line 385 – add "a" after such.

**RESPONSE**: We have added "a" after such.

Line 388-389 - revise sentence to "A substantial number of inundated NWI wetlands were found to coalesce with adjacent wetlands. . .."

**RESPONSE**: We have revised the sentence as suggested.

Line 402 – Do you mean you "could" use it if a dry-period LiDAR was available?

**RESPONSE**: Yes, this is what we meant. We have changed "can" to "could".

Line 384-406 – This is a good discussion of an important issue but it is not entirely clear how this issue affected your findings in this case. I would guess that you likely under-estimated the number of depressions that coalesced?

**RESPONSE**: We thank the reviewer for the encouraging comment. We have to admit that our findings were inevitably affected by the LiDAR data used in this study. Since the LiDAR data were acquired during an extreme wet period, many wetlands already coalesced with adjacent wetlands to form larger wetland complexes. Therefore, it is not possible delineate the individual wetlands before they coalesced unless we had another LiDAR dataset acquired during a dry period. Our methods focused on the potential coalescence between adjacent wetlands when water levels in wetlands continued to increase rather than the coalescence that already took place. In an ideal situation, i.e., the LiDAR data is acquired during an extremely dry period, our methods can simulate the filling-merging-spilling processes and project the coalescences between wetlands. The results can then be validated using the wet-period LiDAR data and aerial photographs.

Line 404-405 – As far as I can tell, however, in this case you did not use the time-series or wet inundation to evaluate or summarize fill-and-spill patterns. Is this correct?

**RESPONSE**: It is correct that we were not able to use the wet-period LiDAR data to evaluate the fill-and-spill patterns. This is partly due to the limitation of the LiDAR data. As noted in the paper, the LiDAR data used in this study were acquired in late October 2011, which was an extremely wet period according to the Palmer Hydrological Drought Index. During such a wet period, a substantial number of inundated wetlands were found to coalesce with adjacent wetlands and form larger wetland complexes. As a result, we were not able to obtain the information of basin morphology of individual depressions before they merged into large wetland complexes. Ideally, using multiple LiDAR datasets acquired in both dry and deluge conditions in conjunction with time-series aerial photographs would be essential for studying the fill-and-spill mechanism of prairie wetlands. In this case, we could use the dry-period LiDAR data to delineate and characterize the morphology of individual wetland depressions before the fill-and-spill processes occur. Furthermore, we can derive the potential flow paths and project the coalescing of wetland depressions after the fill-and-spill processes initiate. The wet-period LiDAR data and time-series aerial photographs can serve as validation datasets to evaluate the fill-and-spill patterns. We plan to further investigate this issue when a dry-period LiDAR data for our study area becomes available.

Line 425 – Can't use the word "accurately" if no validation was done.

**RESPONSE**: We have removed the word "accurately".

Line 433 – Add "potential" before hydrological connectivity.

**RESPONSE**: We have added the word "potential" before hydrological connectivity.

Line 435 – I am struggling with this statement which is used several times throughout the manuscript. Although temporary or seasonal flow paths were likely identified, flowpaths were also likely identified that never actually carry water. How can we distinguish between these or can we?

**RESPONSE**: We thank the reviewer for this concern. We have made changes throughout the manuscript and made it clear that all flow paths identified in this study are potential flow paths. By examining the potential flow paths overlaid on the color infrared aerial photograph, we found that many potential flow paths appeared to be collocated with vegetated areas (see Fig. 9(b)). This indicates that flow paths are likely located in high soil moisture areas that are directly or indirectly related to surface water or groundwater connectivity. We agree with the reviewer that some potential flow paths might never actually carry water. We plan to further investigate this issue in a follow-up study to categorize and validate potential flow paths.

Line 439 –Add what the specific limiting factors have been with traditional remote sensing methods.

**RESPONSE**:  We have modified the sentence as suggested: "Broad-scale prairie wetland hydrology has been difficult to study with traditional remote sensing methods using multi-spectral satellite data due to the limited spatial resolution and the interference of tree canopy (Klemas, 2011; Gallant, 2015)."

1) Klemas, V.: Remote sensing of wetlands: case studies comparing practical techniques, Journal of Coastal Research, 27, 418-427, 2011.
2) Gallant, A.: The Challenges of Remote Monitoring of Wetlands, Remote Sensing, 7, 10938, 2015.

Table 1 – remove extra spaces between Freshwater and Emergent.

**RESPONSE**: We have removed the extra spaces.

Figure 5, 8 and 10 – I would add a basic color to the histograms, maybe light gray? To improve the aesthetics.

**RESPONSE**: We have modified all histograms as suggested.

Figure 6 – Modify x-axis to just show year

**RESPONSE**: We have modified the x-axis as suggested.

Figure 7 – the yellow and blue lines are hard to see, maybe making them a little thicker might make them more visible.

**RESPONSE**: We have made the lines thicker. In addition, we switched the line colors to make them consistent with those shown in Figure 9. Yellow line and blue line represent NWI wetlands and LiDAR-derived inundation areas, respectively.

Figure 10 – This figure gets at several questions I had. Was connectivity calculated so that all wetlands connected to each other and eventually to a stream? And this is then the length distribution of those flowpath lines? If so it should be indicated that these are potential connectivity. What does connected wetlands mean here? Are these just the coalesced wetlands?

**RESPONSE**: We have modified the figure and corresponding text to indicate the potential wetland connectivity. Figure 10a shows the distribution of potential flow path lengths. Figure 10b shows the distribution of elevation differences between wetlands connected through the potential flow paths.

**Our Response to Anonymous Referee #2**

**General Comments**

This manuscript was well thought out, well organized and well written. In the United States the regulatory status of wetlands is currently linked to connectivity to streams so the topic of this manuscript is important. The conceptual model presented for wetland fill and spill seems very useful. The approach used in the reported study is sound and findings support the conclusions reached.

**RESPONSE**: We thank the reviewer for the encouraging comments.

**Specific comments:**

The last paragraph of the introduction is a summary of the study findings. It should be modified to reflect study goals instead.

**RESPONSE**: We thank the reviewer for the good suggestion. We have revised this paragraph and made our research objectives more clear.

"In this paper, we present a semi-automated framework for delineating nested hierarchical wetland depressions and their corresponding catchments as well as simulating wetland connectivity using high-resolution LiDAR data. Our goal was to demonstrate a method to characterize fill-spill wetland hydrology and map potential hydrological connections between wetlands and stream networks. The hierarchical structure of wetland depressions and catchments was identified and quantified using a localized contour tree method (Wu et al., 2015). The potential hydrologic connectivity between wetlands and streams was characterized using the least-cost path algorithm. We also utilized high-resolution LiDAR intensity data to delineate wetland inundation areas, which were compared against the National Wetlands Inventory (NWI) to demonstrate the hydrological dynamics of prairie wetlands. Our ultimate goal is to build on our proposed framework to improve overland flow simulation and hydrologic connectivity analysis, which subsequently may improve the understanding of wetland hydrological dynamics at watershed scales."

Flow routing was performed using D8 algorithm (line 213) but often it has been found that D-infinity algorithms provide more realistic flow characteristics.

**RESPONSE**: We agree with the reviewer that D-infinity algorithms might provide more realistic flow characteristics. In our study, the flow direction raster was generated and used as an intermediate dataset to derive wetland catchments. For delineating catchments/watersheds, we tried the ArcGIS Hydrology Toolbox (https://goo.gl/GhmFld) and the open-source Whitebox Geospatial Analysis Tools (https://goo.gl/dqV4cE). Both software packages use D8 algorithm for watershed delineation. Since our data processing flow was built on the ArcGIS Hydrology Toolbox, for the sake of simplicity, we used the D8 algorithm available in ArcGIS to derive flow directions. Nevertheless, we believe that both flow direction algorithms should lead to the same watershed delineation results.

When reporting numerical results consider the errors associated with the underlying model used to produce the values. The number of nonzero digits should generally reflect the uncertainty. For example see lines 347 and 348 with values reported with 4 significant figures whereas it is known that these estimates have substantial uncertainty. Also in tables with data reported with up to 8 significant digits (Tables 1 to 4).

**RESPONSE**: We appreciate this concern. In the revised manuscript, we have reduced the number of significant digits to no more than three throughout the manuscript.

Figure 7 needs to be reworked. Labels on figure are very difficult to read

**RESPONSE**: We have revised Figure 7. We made the lines thicker. In addition, we switched the line colors to make them consistent with those shown in Figure 9. Yellow line and blue line represent NWI wetlands and LiDAR-derived inundation areas, respectively.

[revised manuscript text omitted]

---

## Author Response (AR2)

**Our Response to Anonymous Referee #1 (second round)**

Comments:

The authors did a nice job responding to the first round of comments. I have two very minor comments and after that I support the article being accepted for publication.

**RESPONSE**: We thank the reviewer again for his/her thorough review and very helpful comments/suggestions.

Comment – Section 3.4 – the authors added the assumptions as requested, but please add an additional comment and reference noting that assuming no infiltration is a reasonable assumption for this landscape. You may want to note that this assumption might be problematic in other landscapes with more heterogeneity in infiltration capacity.

**RESPONSE**:  We have added the additional comment and references as suggested.

"Note that assuming no infiltration is a reasonable assumption for the prairie pothole landscape (Shaw et al., 2013; Hayashi et al., 2016). However, this assumption might be problematic in other landscapes with more heterogeneity in infiltration capacity."

Line 119 – add the actual PHDI value for the wet October.

**RESPONSE**:  We have added the PHDI value.

[revised manuscript text omitted]

---

## Author Response (AR3)

**Our Response to Editor's comments**

Comments:

Thanks for your careful revisions. As one final remark, I would like to make you aware of one recent paper, which might be useful to refer to:

- Blume, T. and van Meerveld, H.J. (2015), From hillslope to stream: methods to investigate subsurface connectivity. WIREs Water, 2: 177–198. doi:10.1002/wat2.1071

However, I leave the decision on this to you.

**RESPONSE**: We thank the Editor for bringing the useful reference to our attention. We have now cited the recommended reference in our manuscript.

Line 444: *"The connectivity between surface and subsurface waters and the associated hydrologic and ecological functions are spatially variable and temporally dynamic (Blume and van Meerveld, 2015)."*